# Ethylene Induction of Non-Enzymatic Metabolic Antioxidants in *Matricaria chamomilla*

**DOI:** 10.3390/molecules25235720

**Published:** 2020-12-03

**Authors:** Veronika Petrulova, Maria Vilkova, Zuzana Kovalikova, Matus Sajko, Miroslav Repcak

**Affiliations:** 1Institute of Biology and Ecology, Faculty of Science, Pavol Jozef Safarik University, Manesova 23, SK-04001 Kosice, Slovakia; veronika.petrulova@upjs.sk (V.P.); sajkomat@gmail.com (M.S.); miroslav.repcak@upjs.sk (M.R.); 2Institute of Chemistry, Faculty of Science, Pavol Jozef Safarik University, Moyzesova 11, SK-04001 Kosice, Slovakia; maria.vilkova@upjs.sk; 3Department of Biology, Faculty of Science, University of Hradec Kralove, Rokitanskeho 62, 50003 Hradec Kralove, Czech Republic

**Keywords:** antioxidants, chlorogenic acids, ethylene, chamomile

## Abstract

Phytochemical investigations of *Matricaria chamomilla* L. (Asteraceae) stated the presence of several compounds with an established therapeutic and antioxidant potential. The chamomile non-enzymatic antioxidant system includes low molecular mass compounds, mainly polyphenols such as cinnamic, hydroxybenzoic and chlorogenic acids, flavonoids and coumarins. The objective of this work was to evaluate the role of the non-enzymatic antioxidant system after stimulation by ethylene in tetraploid chamomile plants. Seven days of ethylene treatment significantly increased the activity of phenylalanine ammonia-lyase, which influenced the biosynthesis of protective polyphenols in the first step of their biosynthetic pathway. Subsequently, considerable enhanced levels of phenolic metabolites with a substantial antioxidant effect (syringic, vanillic and caffeic acid, 1,5-dicaffeoylquinic acid, quercetin, luteolin, daphnin, and herniarin) were determined by HPLC-DAD-MS. The minimal information on the chlorogenic acids function in chamomile led to the isolation and identification of 5-*O*-feruloylquinic acid. It is accumulated during normal conditions, but after the excessive effect of abiotic stress, its level significantly decreases and levels of other caffeoylquinic acids enhance. Our results suggest that ethephon may act as a stimulant of the production of pharmaceutically important non-enzymatic antioxidants in chamomile leaves and thus, lead to an overall change in phytochemical content and therapeutic effects of chamomile plants, as well.

## 1. Introduction

In plants, the non-physiologic over-accumulation of reactive oxygen species (ROS) is related to the effect of biotic or abiotic stimuli. Simultaneously, other physiological defence mechanisms are triggered, which help the normal functioning of plants during stressful conditions. Apart from the enzymatic antioxidants (e.g., superoxide dismutase, peroxidase, or catalase) the non-enzymatic antioxidant system, which includes low molecular mass compounds such as polyphenols (phenolic acids, flavonoids, anthocyanins, lignans, and stilbenes), carotenoids and vitamins (vitamins C and E) is activated, as well [1,2].

Previous phytochemical investigations of *Matricaria chamomilla* stated several compounds with an established therapeutic and antioxidant potential. The chamomile essential oil is characterized by a lower content of polyphenols than the alcohol extracts (methanol or ethanol), so it proves a lower antioxidant activity [3,4]. The main constituents present in alcohol extracts are flavonoids (apigenin, luteolin, quercetin and related derivatives), dicycloethers [5], phenolic acids and their derivatives, e.g., 2-β-d-glucopyranosyloxy-4-methoxycinnamic acids (GMCA) [6], chlorogenic acid derivatives [7], and coumarins [8,9].

From the group of chlorogenic acids, feruloylquinic and mono-, di-, and tri-caffeoylquinic derivatives have been postulated [10]. Information dealing with their specific role in chamomile plants is low, due to the missing detailed identification in plant extracts and subsequent monitoring of changes in their content during induced physiological reactions. The connection of caffeoylquinic acid derivatives in plant response to abiotic stimuli was described, e.g., the effect of short-term UV-B radiation [11] or nutrient imbalance [12]. Thanks to their polyphenol character, chlorogenic acid and its derivatives have the antioxidative potential [13], so they can participate in the non-enzymatic detoxification of oxygen radicals and lipid peroxidation products.

Ethylene is a plant hormone regulating the growth, development, and aging of plants. ROS-generated reagents are active during stress conditions, e.g., cold, salinity, or UV-B radiation. They induce the transcription of genes participating in the ethylene signaling pathway (such as ERF1 as well as LeEILs). Moreover, the ethylene-inducible production of transcription factor ERF1 modulates plant tolerance to abiotic stress, which enhances the transcription of antioxidant enzymes genes coding catalase or glutathione peroxidase [14,15].

The aim of this paper was to evaluate the effect of ethylene on the protective non-enzymatic system of tetraploid chamomile plants, including the accumulation of essential antioxidants, such as flavonoids, cinnamic acids derivatives, and coumarin-like compounds. Based on the ethephon induced changes in the accumulation of chlorogenic acid derivatives and due to the detailed identification of 5-*O*-feruloylquinic acid (5-FQA) isolated from chamomile, offered in this paper, we could determine the role of individual chlorogenic acids in chamomile plant physiology, which had been evaluated at minimum up to now. A comparison of their antioxidative activity and evaluation of changes in their ethylene-induced accumulation can support a hypothesis that 5-FQA represents a storage form of chlorogenic acid derivatives in chamomile plants. In addition, an evaluation of the major chemical constituents by quantum chemical studies (DFT B3LYP/3-31G**), such as maps of molecular electrostatic potential (MEP), frontier orbital’s (HOMO and LUMO) of chlorogenic acid, 1,5-DCQA, and 5-FQA was conducted. The general scheme summarizing the methodological steps is presented in Figure 1.

## 2. Results and Discussion

Ethephon (ET) acts as a plant growth regulator converted into ethylene in plant metabolism. Ethylene is involved in most plant defensive responses induced by abiotic stress [16,17] or wounding [18]. Its elevated production affects many physiological processes in plants.

We mainly focused on the evaluation of the non-enzymatic antioxidant metabolite system, which occurred in leaves treated by 0.01% ET solution in triple spraying. Previously published results [19] using the same ET concentration point to the significant changes in the accumulation of several coumarins, total soluble phenols, and flavonoids. However, the work does not specifically address phenolic acids or flavonoids, which are more effective antioxidants. For this reason, we chose the given concentration and focused in detail on the selected metabolites. The HPLC-DAD-MS analysis of chamomile leaf methanol extract determined an induced accumulation of flavonoids aglycons, simple cinnamic acids, chlorogenic acid derivatives, and some coumarins with their related compounds (Figure 2 and Figure 3).

### 2.1. Effect of Ethephon on PAL Activity and Phenylpropanoids Accumulation

All the evaluated components of the non-enzymatic antioxidant system are related to the phenylpropanoid biosynthetic pathway, in which phenylalanine ammonia-lyase (PAL) has a crucial position [20]. In our work, the used ET treatment stimulated the PAL activity in leaves (Figure 4). A correlation of PAL activity and ethylene has been already described in the work of Heredia et al. [21], in which the applied ethylene stimulated an increase in both the enzyme activity and antioxidants content, specifically 1,5-dicaffeoylquinic acid (1,5–DCQA), ferulic acid, and isocoumarin.

One of the first products of the phenylpropanoid pathway after the formation of cinnamic acid are the active forms of its derivatives, such as *p*-coumaric, caffeic, sinapic, or ferulic acid [22]. Our results in Table 1 show an increased production of the mentioned phenolic acids, already 24 h after the last treatment, except for the *p*-coumaric acid. The most significant increment was recorded for syringic acid (about 170%). Other substantial elevations were monitored for the vanillic (50%) and caffeic (34%) acid. Ferulic acid acts as a precursor of vanillic and syringic acid [23], so its slightly increased pool may correlate to the significantly elevated levels of both related acids. Both of them expressed significant antioxidative properties, and they are useful as free radical scavengers [24]. The reason for *p*-coumaric acid’s decreased content correlates with its precursor role in the biosynthesis of caffeic acid and its related derivatives (feulic, vanillic and syringic acid), in which the content was enhanced as mentioned above.

In the methanolic extract, mainly inactive glycosides are presented, and related aglycones are the minority. The mutual ratio of glycosides and aglycons amounts indicates the actual physiological state of the plant, since inactive glycosides are considered as a storage form with minimal physiologic effects [6]. Flavonoids in chamomile leaves are usually determined as a sum of flavonols [11,19], while their detection in flowers or foods is more specified. Kováčik and Klejdus [12] offered a more detailed determination of flavonoids in fresh chamomile leaves. The authors established three flavonols (hyperoside, rutin, astragalin) and three flavones glycosides (luteolin-7-*O*-glucoside, luteolin-3′,7-*O*-diglucoside, and apigenin-7-*O*-glucoside) in hydroponically cultivated plants affected by nutrient stress.

From our results, the qualitative and quantitative determination of free quercetin, luteolin, and kaempferol aglycons in ET-treated leaves indicates the mobilization of active forms in conditions simulating stress stimuli. During normal conditions, their levels seemed to be in the following order: Quercetin < kaempferol < luteolin (Figure 5). The accumulation of quercetin and luteolin arose substantially 24 h from the last ET-treatment and simultaneously, the kaempferol level decreased. This state was maintained during the experiment. In general, quercetin is the most abundant flavonoid in plants, acting as an effective protectant against the harmful effects of ROS [25]. Thanks to the dihydroxylated B-ring of luteolin and quercetin, both could be considered as better antioxidant agents than monohydroxylated flavonoid derivatives (such as kaempferol) [26]. Their ET-stimulated production and antioxidant potential destined them for participation in the active non-enzymatic antioxidant system of chamomile.

The used ET treatment had an impact on the accumulation of other main metabolites which occurred in the methanol leaf extract (GMCAs, herniarin and daphnetine glucoside—daphnin), as well (Table 1). Decreased levels of glycosylated cinnamic acid derivatives were in correlation with higher levels of their active forms, such as cinnamic acid and related derivatives with a substantial antioxidant potential. From coumarin-related compounds, daphnin and herniarin were detectable. The production of both after the treatment increased, however, the amount of daphnetin was minimal. In our experiment, umbelliferon was not detected, but its increment in ET-treated chamomile leaves was shown in [19]. The production of free coumarin forms, mainly umbelliferon, is connected with the reaction induced by wounding or severe abiotic stress [6,27]. The biosynthesis of coumarins in plants is still under investigation due to their different biological effects. Plants are natural sources of coumarins, but their low levels and instability during isolation lead to the development of other ways of acquisition. Bourgaud et al. [28] suggested the hydroxylation of umbelliferon to dihydroxycoumarins (e.g., daphnetin). However, it has been just confirmed in in vitro experiments. Similarly, Chu et al. [29] successfully produced three different molecules herniarin, esculetin, and skimmin from umbelliferon in in vitro conditions. Based on the results of the mentioned experiments, an increased level of umbelliferon in ET-treated chamomile leaves [19] could be correlated with increased levels of herniarin and daphnin in our results. Chamomile coumarins showed an antioxidant activity in the DPPH assay [30]. The position and type of substituent attached to the aromatic ring of coumarin molecules have a great influence on the antioxidant activity of these metabolites. The higher number of hydroxyl group on the ring structure of coumarins is correlated with higher ROS suppression effects in comparison with methoxylated or non-substituted coumarins.

### 2.2. Role of Chlorogenic Acid Derivatives in Chamomile Plant Physiology and Their Antioxidant Potential

In normal physiological conditions, an average 10-fold higher accumulation of leaf 5-FQA than chlorogenic acid and 1,5-DCQA was shown (Figure 6). However, 24 h after the ET treatment, the content of 5–FQA significantly decreased with a simultaneous increment in the chlorogenic acid and 1,5-DCQA content. This state was maintained without changes through the next 24 h after the last spraying. Mutual changes of quinic acid derivatives in chamomile have not been described in detail yet. A similar trend in changes of chlorogenic acids accumulation was monitored after the short-term effect of UV-B radiation on chamomile leaves (unpublished data for 5-FQA; Figure A1 and Figure A2 in Appendix B) [11]. Moreover, a significant enhancement of the chlorogenic acid and 1,5-DCQA effect was recorded here. The derivatives of caffeic acid, such as chlorogenic (3-caffeoylquinic acid) and 1,5-DCQA, contain several free hydroxyl groups and express strong antioxidative properties, which were demonstrated in [13,31].

One possible explanation for the enhanced accumulation of caffeoylquinic derivatives in conditions of higher ROS concentration induced by ethephon or UV-B radiation could be their better antioxidant potential in comparison with feruloyl derivatives. Kono et al. [32] declared substantial antioxidant properties of the chlorogenic acid (3-caffeoylquinic acid), and Chan et al. [33] proved the higher effectivity of caffeoylquinic acids to scavenge free radicals than any other phenols. Testing 16 different phenolic compounds, including caffeic and ferulic acids, on the DPPH scavenging activity showed higher effectiveness of caffeic acid to scavenge radicals than ferulic acid [34]. It could be supposed that chlorogenic acids with a higher number of caffeoyl groups could prove a more significant antioxidative capacity and reactivity, and physiological efficacy than feruloyl derivatives. This assumption was demonstrated in the DPPH testing of the antioxidative capacity of standards of chlorogenic acid derivatives confirmed in chamomile plants and isolated 5-FQA (Figure 7). The DPPH assay is widely used to evaluate the free radical scavenging effectiveness, which is the antioxidant activity of various substances. It is expressed as a percentage of reduction of DPPH radicals by testing substances or as an EC50 value, which is the concentration of the extract determining a 50% decrease of absorbance of the DPPH solution [35]. From our results (Figure 7), the highest EC50 value showed 5-FQA, but it reflected the lowest antioxidative capacity (AC) in comparison with the AC of chlorogenic acid and 1,5-DCQA. Yang et al. [36] obtained similar results, which showed a higher AC of metabolites with the caffeoyl group presented in the molecule than these containing only feruloyl. Moreover, they concluded that the antioxidant activity is determined not only by the number of phenolic OH-groups, but also by their position in the aromatic rings. Compounds exhibiting vicinal OH-groups on the aromatic rings, such as dihydrocaffeic acid, are particularly potent antioxidants.

A detailed identification of 5-FQA contributed to the elucidation of the role of chlorogenic acid derivatives in chamomile plant physiology. In chamomile plants, several quinic acids (such as caffeoyl and feruloyl derivatives) were recorded mostly by LC-MS methods using the related chemical standards [7,10]. Up to now, 5-FQA was determined similarly. Here, its LC isolation and following NMR identification was conducted (Figure 8). Its high content in normal physiological conditions and decreased amount after the stress indicate its storage role of chamomile quinic acid derivatives. However, for an absolute confirmation of the hypothesis dealing with other changes of chlorogenic acid derivatives, more detailed analyses of participating enzymatic and molecular components of corresponding chamomile biosynthetic pathways are necessary.

### 2.3. Molecular Modelling

The map of the molecular electrostatic potential (MEP) shows the regions which are specifically coloured depending on the potential/charges. The blue/green colour indicates the positive regions, the parts of the molecule with a higher probability to suffer a nucleophilic attack. The negative regions visualized by a green/red colour indicate the regions which are probably performing a nucleophilic attack [37]. MEPs of the studied quinic derivatives are presented in Appendix A in the Supplementary Material.

The maximum positive electrostatic potential of the compounds is relatively similar: 0.08243 a.u. for chlorogenic acid, 0.08123 a.u. for 5-FQA, and 0.07957 a.u. for 1,5-DCQA. The minimum electrostatic potential varied from −0.08243 to −0.07957 a.u. for 1,5-DCQA. The distribution pattern of the positive and negative potential is uniform; mainly on hydrogens (H) directly bonded to the oxygen atoms (–OH groups) for the positive potential, and on oxygen directly bounded on other atoms or in –OH groups for the negative potential.

The frontier orbitals provide information on the highest occupied molecular orbital (HOMO) energies and the lowest unoccupied molecular orbital (LUMO) energies, in addition to acting as electron-donors and electron-acceptors, respectively. An important electronic parameter for describing the antioxidant ability of the studied molecules is the HOMO energy. Therefore, the higher the HOMO energy, the higher the capacity to realize nucleophilic attacks, i.e., donating electrons and scavenging the radical [37]. 1,5-DCQA is presented by the highest HOMO value, −5.484 eV, followed by 5-FQA (−5.5648 eV) and chlorogenic acid (−5.888 eV). HOMO regions are located mainly over aromatic rings and double bounds of caffeic and ferrulic parts of chlorogenic acid and 5-FQA, respectively. In 1,5-DCQA, HOMO is located only over one aromatic ring of caffeic parts.

On the other hand, the analyses of LUMO orbitals bring information on the region susceptible to suffer nucleophilic attacks. Therefore, the lower the LUMO energy, the lower the resistance to accept electrons. The minimum value was represented by 1,5-DCQA (−1.937 eV). The distribution of LUMO regions is similar to HOMO, over the aromatic rings and double bonds. In 1,5-DCQA, it is mainly located on the aromatic ring of the other caffeic parts.

For each molecule, the gap energy value (ΔE = E_LUMO_ − E_HOMO_) indicating the chemical reactivity and molecular stability was calculated. Molecules with high HOMO values and a low gap value show higher reactivity and consequently less stability, which predetermines them to have an antioxidant activity by scavenging ROS. Here, the highest HOMO and the lowest gap value (3.92 eV) was observed for 5-FQA, although Urbaniak et al. [38] showed better properties of the caffeic acid derivatives than ferulic acid and its derivatives.

Gallic acid represents a molecule with significant antioxidative properties and therefore, it is widely used as a standard compound in many antioxidative assays. A comparison of our results with the data on gallic acid available in [37] can provide information on the degree of antioxidant effects of the investigated quinic derivatives. The HOMO, LUMO, and gap values for gallic acid were 1.2327, 0.2414, and 1.474 eV, respectively. All these values meet the criteria significantly better, which suggests better antioxidant properties compared to the studied substances.

For in silico analyses of the studied quinic acid derivatives, the Molinspiration Cheminformatics and Swiss Target Prediction servers were used. The molecules considered as a potential orally active compound had to satisfy the Lipinski’s rule of five of the drug-like properties. These physicochemical parameters are associated with an acceptable aqueous solubility and intestinal permeability and comprise the first steps in oral bioavailability. For example, the lipophilicity coefficient (LogP) is a property that quantitatively measures the lipophilia of the compounds, one of the most important molecular properties for drug absorption [39]. It was observed that only 5-FQA did not violate the rule of five (Appendix A), i.e., ≤ 5 H-bond donors, ≤ 10 H-bond acceptors, < 500 g mol^−1^ of molecular weight, < 5 logP coefficient, < 10 rotatable bonds, and topological polar surface area (TPSA) < 140. 1,5-DCQA violates three rules due to the higher molecular weight and number of H-bond donors and acceptors. Detailed in silico studies were performed only for the chlorogenic acid (no available data for 5-FQA and 1,5-DCQA), which on the basis of the results, such as identification of its thermodynamically stable conformation or nucleophilic site, appears to be a promising drug [40]. Although the chlorogenic acid violates one rule (number of H-bond donors), its pharmaceutical properties are extensively studied due to the fact that it is the most abundant isomer among caffeoylquinic acid isomers and one of the most available acids among phenolic acid compounds, which can be naturally found in green coffee extracts and tea [41].

The Swiss target prediction estimates the most probable macromolecular targets of the tested molecules, which were assumed as bioactive. In this server, the targets prediction is obtained by the combination of molecular 2D and 3D similarity with a library of 370,000 known actives on more than 3000 proteins from three different species. The Swiss prediction for all three molecules was similar, but the score differed. The highest score was found for two enzymes: Aldose reductase and aldo-keto reductase family 1 member B10, and protease matrix metalloproteinase 2 (see Appendix A).

## 3. Materials and Methods

### 3.1. Plant Material and Treatments

Seeds of tetraploid plants of *Matricaria chamomilla* L. (cv. ‘Lutea’) were germinated in the sand under laboratory conditions. Three-week-old seedlings were transferred into the pot filled with 130 g of garden substrate with a 60% water-holding capacity. They were cultivated at 25 °C with a 12-h photoperiod under fluorescent white light (TLD 36 W/33 Philips fluorescent tubes with a photon flux density 210 µmol m^−2^ s^−1^). After the next 5 weeks, plants were divided into two groups: Controls and treated by ethephon (2-chloroethyl phosphonic acid) (Sigma, Canada). Control plants were chemically untreated, cultivated under the same conditions such as the treated ethephon, and collected in time harvesting of the first treated group. During the 7 days, the 0.01% aq. solution of ethephon was triple sprayed on chamomile leaves and plant material was analyzed 24 and 48 h after the last spraying. Ethephon is usually used as an ethylene releaser due to its gradual degradation to ethylene after its contact with the plant surface.

### 3.2. Isolation and Identification of 5-O-feruloylquinic Acid

An ethanolic extract of leaves (50.2 g) was fractionated in column chromatography. Silica gel (70–140 µm) was deactivated by 11% of water and eluted with mobile phase CHCl_3_ contained from 5 to 20% of methanol. The desired fraction (1.8 g) was re-chromatographed and eluted with 1% of methanol in CHCl_3_. After that, 40 mg of brown compound was obtained and used for NMR analysis. For the separation and purification of 5-FQA, the preparative HPLC with DAD detection with a column Kromasil100 C18 (250 × 4.6 mm, 5 μm) was used (Agilent Technologies 1260 Infinity device, Santa Clara, CA, USA).

The structure of 5-FQA was determined by 1D (^1^H, ^13^C, selective NOESY, TOCSY) and 2D (gCOSY, NOESY, gHSQCAD, gHMBCAD) NMR spectroscopy (Figure 8). The residue was dissolved in deuterated methanol (CD_3_OD) and chemical shifts were referenced to the residual solvent peak (^1^H NMR 3.31 ppm and ^13^C NMR 49.1 ppm). The NMR spectra were recorded on a 600 MHz Varian VNMRS NMR spectrometer (Agilent, Palo Alto, USA, CA) equipped with a 5 mm One NMR probeat at 25 °C. The melting point of the substance was set at 117–121 °C.

The 7-point calibration curve was constructed by analyzing various amounts of 5-FQA (21.57 mg/100 mL diluted in 75% (*v*/*v*) methanol, from which 1, 2, 3, 4, 5, 6, 7 μL were taken and analyzed). Every point of the curve is the result of two-injection analyses. The whole range of tested concentrations showed good linearity with a regression coefficient (R2) better than 0.999. Limits of detection (LOD) and quantification (LOQ) were not determined due to the high metabolite accumulation in plants.

*5-O-feruloylquinic acid*: ^1^H-NMR (600 MHz, CD_3_OD-*d*_4_): δ = 7.58 (1H, d, *J* 15.9 Hz, H-3), 7.08 (1H, d, *J* 2.1 Hz, H-2′), 7.06 (1H, dd, *J* 8.3, 2.1 Hz, H-6′), 6.94 (1H, d, *J* 8.3 Hz, H-5′), 6.32 (1H, d, *J* 15.9 Hz, H-2), 5.34 (1H, td, *J* 9.0, 4.4 Hz, H-3′’), 4.18 (1H, m, H-5′’), 3.89 (3H, s, OCH3), 3.73 (1H, dd, *J* 8.4, 3.0 Hz, H-4′’), 2.23 (1H, d, *J* 11.4 Hz, H-2′’a), 2.18 (1H, dd, *J* 14.2, 3.3 Hz, H-6′’a), 2.06 (2H, m, H-2′’b, H-6′’b). ^13^C-NMR (150 MHz, CD_3_OD-*d*_4_): δ = 177.2 (COOH), 168.5 (C-1), 151.5 (C-4′), 147.9 (C-3′), 146.7 (C-3), 128.9 (C-1′), 122.9 (C-6′), 116.3 (C-2), 114.8 (C-2′), 112.5 (C-5′), 76.3 (C-1′’), 73.5 (C-4′’), 72.0 (C-3′’), 71.4 (C-5′’), 56.4 (OCH3), 38.8 (C-2′’), 38.2 (C-6′’).

### 3.3. Extraction and HPLC Analyses of Secondary Metabolites and PAL Activity

For the analysis of main chamomile chlorogenic acids, 100 mg of dry leaves were extracted with 5 ml of 75% methanol. Extracts were analyzed by the semipreparative HPLC with DAD detection (Agilent Technologies 1260 Infinity device) using a column Kromasil 100 C18 (250 × 4.6 mm, 5 μm). Quantitative data were collected using the Chemstation software rev. B.04.03 [16] (Agilent, Santa Clara, CA, USA). Mobile phases A (acetonitrile:trifluoroacetic acid:H_2_O; 10:1:89) and B (acetonitrile:H_2_O; 70:30) were in a gradient program with a flow rate of 0.7 ml min^−1^: 0–25 min 50% B; 25–35 min 100% B; 35–45 min 100% B; 50 min 10% B. The single inject volume was 20 μL and the wavelength of detection was 320 nm.

The HPLC solvents such as acetonitrile (Sigma-Aldrich, Darmstadt, Germany), trifluoroacetic acid (Sigma-Aldrich, Steinheim, Germany), methanol (Sigma-Aldrich, Steinheim, Germany), and deionized water (Millipore Direct-Q 3UV with pump, Merck, Darmstadt, Germany) were used. Herniarin (≥99%) was purchased from Extrasynthése (Genay Cedex, France). The chlorogenic acid (≥95%) and 1,5-dicaffeoylquinic acid (≥98%) were purchased from Fluka. (*Z*)- and (*E*)-GMCA and daphnin were isolated from a chamomile leaf extract and their purity was assessed by HPLC-DAD and NMR [8,42].

The activity of phenylalanine ammonia-lyase (PAL, EC 4.3.1.5) was determined as the production of *trans*-cinnamic acid (*t*-CA) from phenylalanine using the HPLC method [43] in homogenates prepared using a sodium borate buffer (pH 8.8). The HPLC conditions: Kromasil SGX C18 7 µm (150 × 4.6 mm) column, mobile phase was 55% acetonitrile in the flow rate of 0.8 ml min^−1^. The detection of *t*-CA was carried out by the retention time and performed at 275 nm. The activity of PAL was expressed as nmol *t*-CA min^−1^ g^−1^ fresh wt. The *t*-CA (Sigma-Aldrich, Steinheim, Germany) standard compound was used for the quantification.

### 3.4. HPLC-MS Analysis of Phenolic Acids and Flavonoids

The MS analysis of phenolic acids and flavonoids was performed on UHPLC on a 2.1 × 50 mm, 1.8 µm Zorbax RRHD Eclipse plus C18 column (Agilent) with a 6470 Series Triple Quadrupole mass spectrometer (Agilent) (electrospray ionization—negative ion mode) as a detector (Santa Clara, CA, USA). The source parameters are reported in Table 2. Eluents: (A) 0.05% formic acid in water and (B) 0.05% formic acid in acetonitrile were used in the following gradient program: 0–1 min (5% B), 2.0–4.0 min (20% B), 8.0–9.5 min (70% B), 10.0–11.0 min (5% B). Selected MRM transitions were followed for each compound: Caffeic acid (179.0 = > 135.0, 107.0), vanillic acid (167.0 = > 152.0, 108.0), syringic acid (197.1 = >182.0, 123.0), *p*-coumaric acid (163.1 = > 119.0, 104.9), ferulic acid (193.1 = > 134.1, 178.0), chlorogenic acid (353.1 = > 191.0, 127.0), kaempferol (185.1 = > 1185.0, 239.0), luteolin (285.1 = > 133.0, 151.0), quercetin (301.0 = > 151.0, 179.0) [44]. Caffeic acid (≥ 98%), vanillic acid (≥ 97%), syringic acid (≥ 95%), *p*-coumaric acid (≥ 98%), ferulic acid (≥ 99%), and quercetin (≥ 95%) were purchased from Sigma-Aldrich (Steinheim, Germany). Kaempferol (≥ 97%) and luteolin (≥ 98%) were purchased from Extrasynthése (Genay Cedex, France).

### 3.5. Antioxidant Capacity Assay

The antioxidant capacity (AC) of quinic acid derivatives, 5-FQA, chlorogenic acid, and 1,5-dicaffeoylquinic acid (1,5-DCQA), was assayed by measuring the decrease in absorbance at 515 nm of the stable free radical DPPH [34,45]. The purple-coloured free radical reacts with the scavenger to yield the colourless product 1,1-diphenyl-2-picrylhydrazine. From the quinic acid derivatives stock solutions (1 mg of chlorogenic and 1,5-DCQA in 2 mL of 75% methanol; 1.43 mg of 5-FQA in 2 mL of 75% methanol) 100, 50, 25, 12.5, 6.25, 3.125, 1.075 μL were taken and mixed with 100 μL 0.2 mM DPPH. The absorption was recorded after 30 min on the fluorescent reader (Synergy HtBiotek, Bad Friedrichshall, Germany). AC was determined as the percentage of radical scavenging calculated according to the following equation:[%] = 100 × (Ac − As)/Ac

Where As is the absorbance of the sample and Ac is the absorbance of the control. The percentage of DPPH scavenging versus the concentration of the sample was plotted. The concentration of the sample necessary to decrease the DPPH concentration by 50% was obtained by interpolation from the linear regression analysis and denoted a EC50 value (μg mL^−1^). All determinations were carried out in triplicate.

### 3.6. Molecular Modelling

Molecular modelling started with the construction of chlorogenic acid, 1,5-DCQA, and 5-FQA in the program GaussView 6.0 [46], followed with computational calculations in the Gaussian 16 program [47]. The compounds were optimized in the density functional theory method (DFT) in theory level B3LYP/6-31G** and frequencies. The constructions of the MEP and the frontier orbitals (HOMO and LUMO) were visualized with the aid of the Avogadro program [48].

The Molinspiration Cheminformatics and Swiss Target Prediction (Lausanne, Swiss) servers were used for determination of in silico predictions.

### 3.7. Statistics

The obtained data were statistically analyzed using ANOVA and Tukey’s test in Minitab Release 11 (Minitab Inc., State College, PA, USA). The number of replications (n) in the figures denotes individual plants measured for each parameter.

## 4. Conclusions

In summary, the application of ethephon (as a source of ethylene) on chamomile plants influenced their non-enzymatic antioxidant system involving compounds of the phenylpropanoid pathway with high antioxidant potentials, such as hydroxycinnamic, hydroxybenzoic and chlorogenic acids, flavonoids, and coumarins. Using ethylene could be understood as a way to change the phytochemical content and to ameliorate the therapeutic effects of chamomile plants. Moreover, this paper indicated the functions of chamomile chlorogenic acid derivatives. Each of the other changes in chlorogenic acid derivatives levels induced by the abiotic factor has shown a need for a more detailed study of the activity participating in enzymatic and non-enzymatic components of this branch of phenylpropanoid pathway. However, this should be preceded by a molecular analysis and identification. Our results indicated the assumed antioxidant properties of chlorogenic acid derivatives, which are based on molecular modelling including the analysis of MEP, frontiers orbitals (HOMO, LUMO), and in silico predictions using the Swiss Target Prediction and the Molinspiration servers. However, a more detailed determination of antioxidant properties in the sense of violating a rule for oral bioavailability, log P, mainly for 5-FQA and 1,5-DCQA, requires more precise ways of modelling.

## Figures and Tables

**Figure 1 molecules-25-05720-f001:**
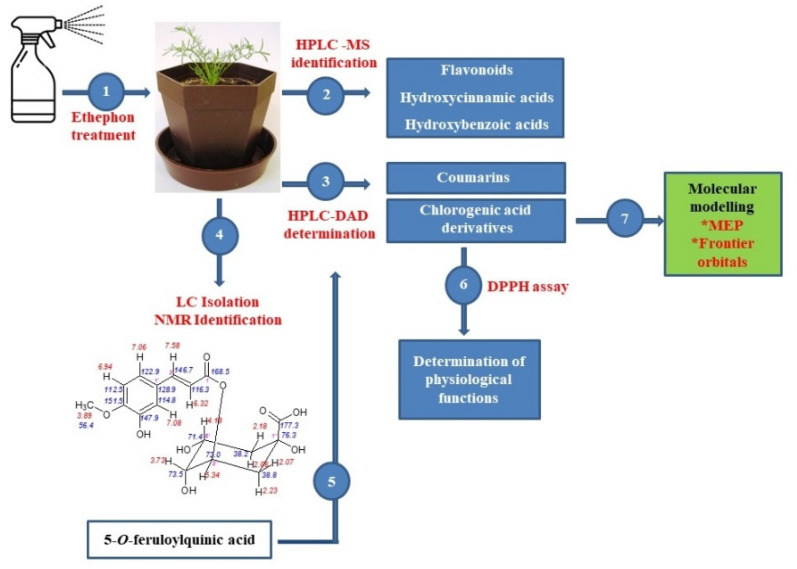
General scheme summarizing the methodological steps.

**Figure 2 molecules-25-05720-f002:**
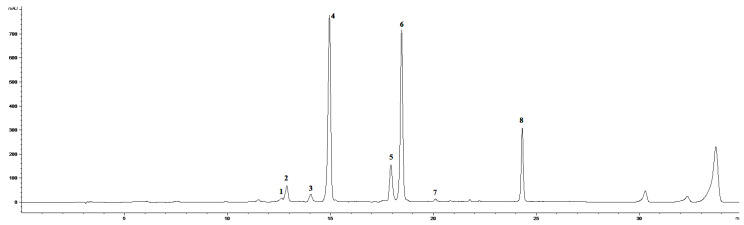
HPLC-DAD chromatogram of methanolic extract of *Matricaria chamomilla* leaves (320 nm) in time 0 h. (**1**) Skimmin, (**2**) daphnin, (**3**) chlorogenic acid, (**4**) *Z*-GMCA, (**5**) 5-FQA, (**6**) *E*-GMCA, (**7**) 1,5-DCQA, (**8**) herniarin.

**Figure 3 molecules-25-05720-f003:**
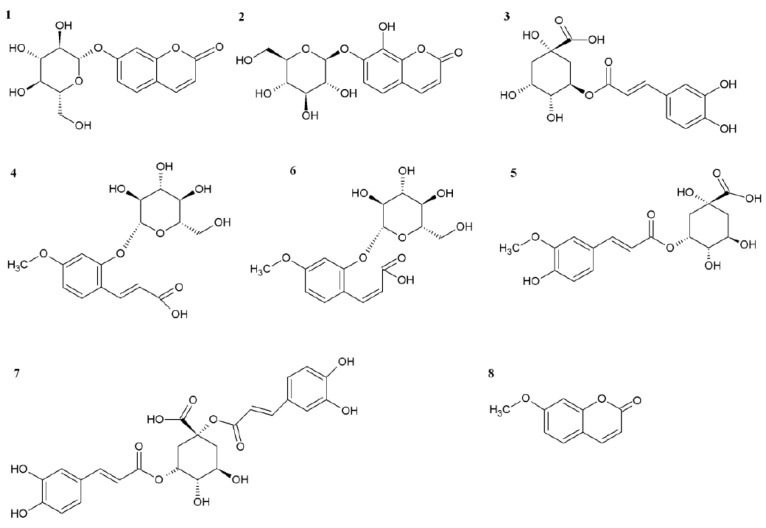
Chemical structures of molecules shown in the HPLC-DAD chromatogram.

**Figure 4 molecules-25-05720-f004:**
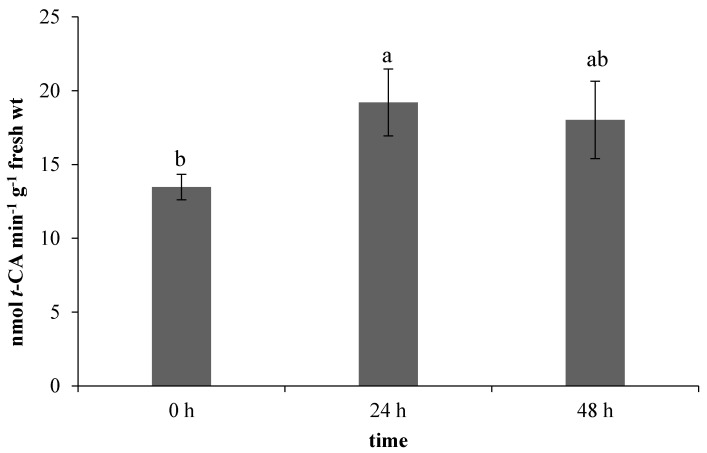
Effect of ethephon on phenylalanine ammonia-lyase (PAL) activity (nmol *t*-CA min^−1^ g^−1^ fresh wt) in *Matricaria chamomilla* leaves. Data are means ± SDs (*n* = 3). Values within columns, followed by the same letter(s), are not significantly different according to Tukey’s test (*p* < 0.05).

**Figure 5 molecules-25-05720-f005:**
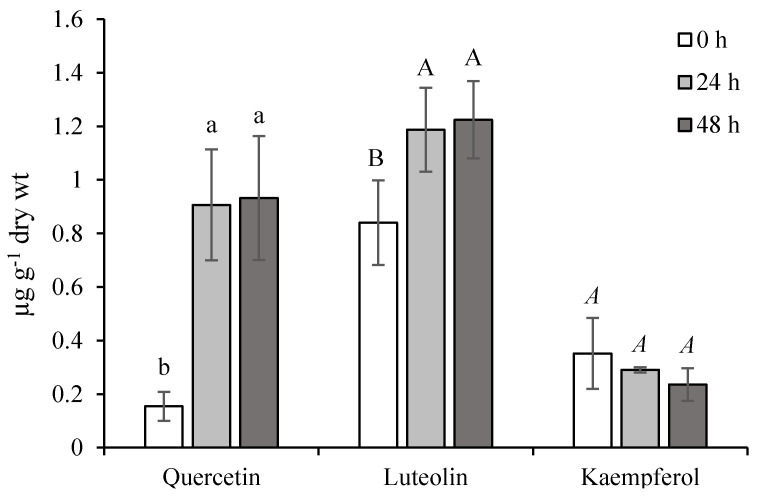
Effect of ethephon on flavonoids accumulation (µg g^−1^ dry wt) in *Matricaria chamomilla* leaves. Data are means ± SDs (*n* = 3). Values within columns, followed by the same letter(s), are not significantly different according to Tukey’s test (*p* < 0.05) The same type of letters shows a significant difference between the values of one compound.

**Figure 6 molecules-25-05720-f006:**
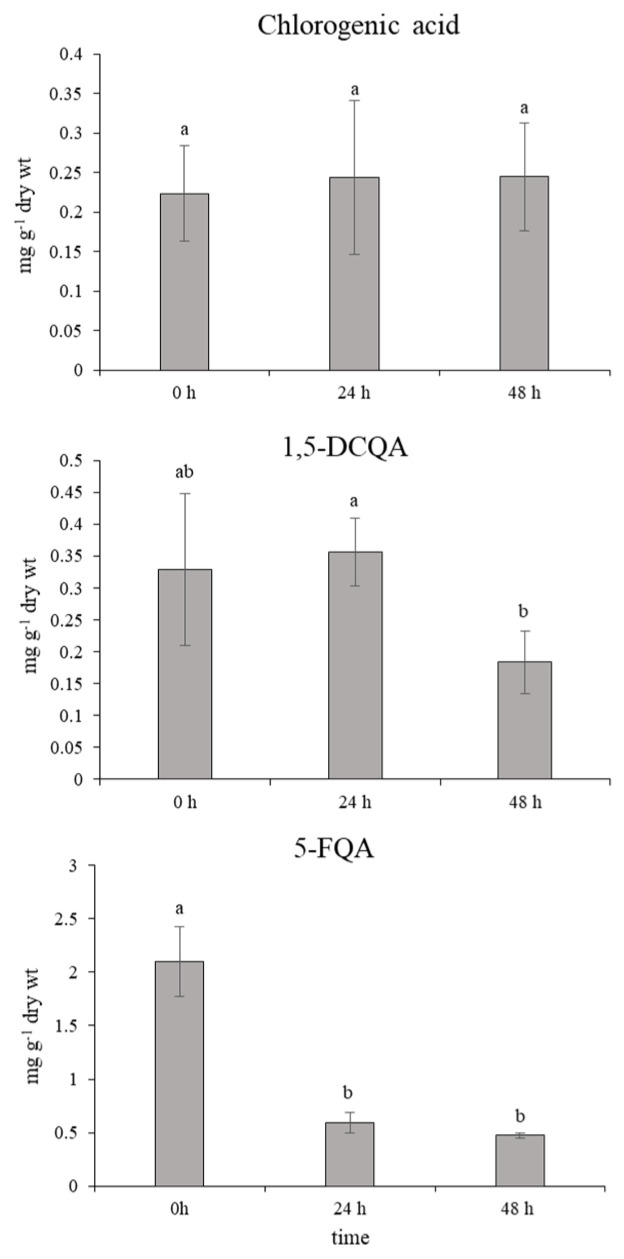
Effect of ethephon on quinic acid derivatives accumulation (mg g^−1^ dry wt) in *Matricaria chamomilla* leaves. Data are means ± SDs (*n* = 5). Values within columns, followed by the same letter(s), are not significantly different according to Tukey’s test (*p* < 0.05).

**Figure 7 molecules-25-05720-f007:**
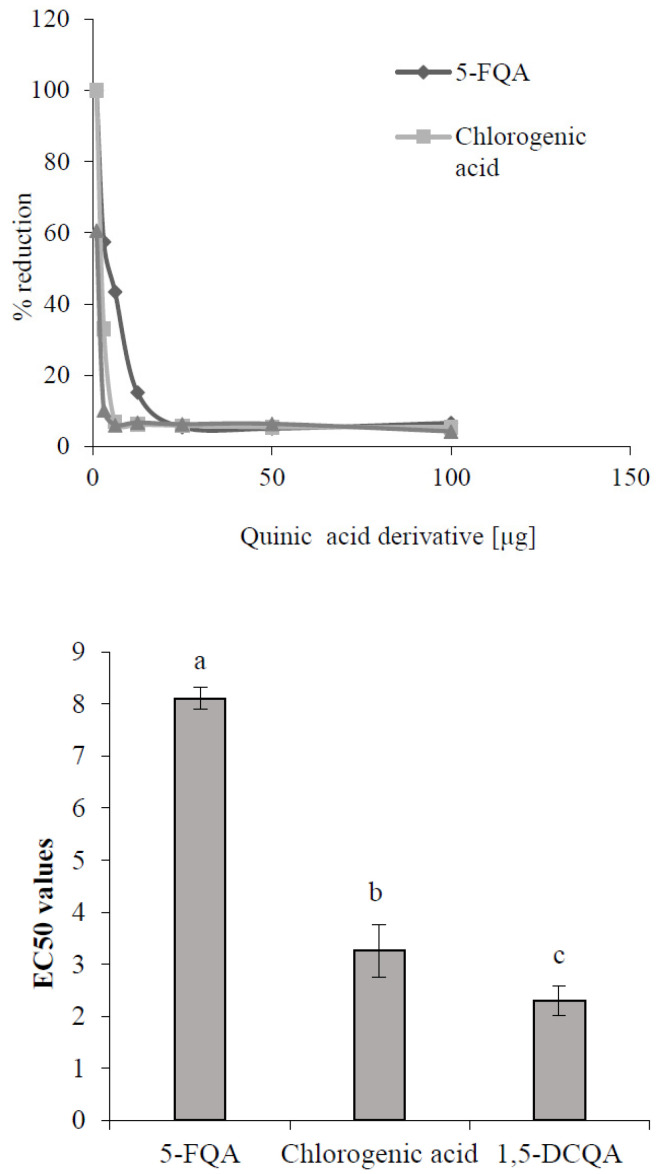
Antioxidant capacity (**up**) and EC50 values (**down**) of selected quinic acid derivatives determined by the method of 2,2-diphenyl-1-picrylhydrazil (DPPH) free radical scavenging. Each value is the mean ± SD (*n* = 3). Different letters above columns means significant difference of values according to Tukey’s test (*p* < 0.05).

**Figure 8 molecules-25-05720-f008:**
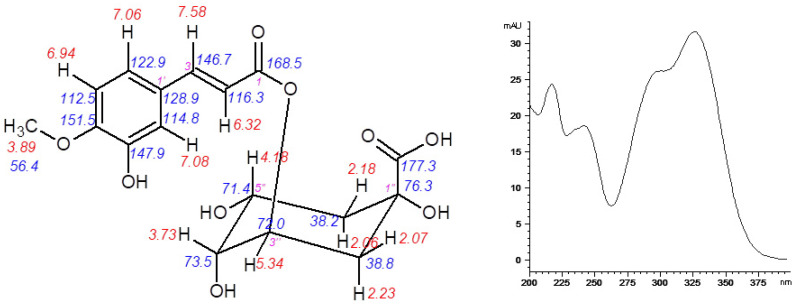
Structure of the isolated 5-*O*-feruloylquinic acid together with ^1^H and ^13^C-NMR chemical shifts (**left**) and UV spectrum (**right**).

**Table 1 molecules-25-05720-t001:** Effect of ethephon on simple phenolic acids levels and major native secondary metabolites in *Matricaria chamomilla* leaves. Data are means ± SDs (*n* = 5). Values within lines, followed by the same letter(s), are not significantly different according to Tukey’s test (*p* < 0.05).

	Ethephon Treatment		
	0 h	24 h	48 h
Hydroxycinnamic and hydroxybenzoic acid derivatives (µg g^−1^ dry wt)			
Caffeic acid	7.445 ± 0.595 ^b^	10.047 ± 0.964 ^b^	14.149 ± 2.399 ^a^
Vanillic acid	41.368 ± 6.236 ^b^	62.463 ± 6.877 ^a^	29.276 ± 6.914 ^b^
Syringic acid	1.026 ± 0.601 ^bc^	2.834 ± 0.5633 ^ab^	4.780 ± 1.372 ^ab^
*p*-Coumaric acid	1.064 ± 0.267 ^ab^	0.592 ± 0.107 ^b^	1.596 ± 0.526 ^a^
Ferulic acid	0.485 ± 0.070 ^a^	0.504 ± 0.1771 ^a^	0.692 ± 0.281 ^a^
Coumarin-related compounds (mg g^−1^ dry wt)			
Daphnin	0.950 ± 0.193 ^b^	1.626 ± 0.113 ^a^	1.125 ± 0.146 ^b^
*Z*-GMCA	6.602 ± 0.849 ^a^	1.225 ± 0.390 ^b^	6.032 ± 0.539 ^a^
*E*-GMCA	9.626 ± 0.71 ^a^	6.867 ± 0.686 ^b^	7.455 ± 1.307 ^ab^
Herniarin	0.107 ± 0.0381 ^b^	0.236 ± 0.064 ^a^	0.259 ± 0.023 ^a^

**Table 2 molecules-25-05720-t002:** HPLC-MS source parameters.

Parameters	Value
Gas temperature	300 °C
Gas flow	9 L min^−1^
Nebulizer	35 psi
Sheat gas temperature	380 °C
Sheat gas flow	12 L min^−1^
Capillary voltage	5000 V
Nozzle voltage	300 V

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
