# Peer review of "Ethylene Induction of Non-Enzymatic Metabolic Antioxidants in Matricaria chamomilla"

_molecules, 2020, doi:10.3390/molecules25235720_

Round 1
Reviewer 1 Report
The Manuscript provided by Petrulova V, describes the investigation of antioxidant components in Matricaria chamomilla. The instrumentation is state-of-the-art and the article is suitable for Molecules.
However, there are some flaws surrounding this piece of work.
While the text is well written, the figures look poor and in many cases are difficult to read and the labels on the axis are missing. I suggest polishing the figures and removing the black frame.
Secondary I also suggest to provide a figure with the chemical structure of the main identified compounds. This will help to guide the readers trough the manuscript.
As minor point, I would suggest to provide stars on the bar chart for significance.
It is unclear why and where MS was used in the manuscript. Was it used only for identification prior to NMR? This section is briefly mentioned in the material and methods I would kind ask to provide more details.
Additionally some parameters such as, spray voltage, mass isolation, calibrated lenses, declustering potential, are missing. I suggest to provide a table with this parameters in the supplementary material.
Author Response
We thank Reviewer for incitement comments and comments. All suggested necessary changes were considered and are included in the manuscript.
- While the text is well written, the figures look poor and in many cases are difficult to read and the labels on the axis are missing. I suggest polishing the figures and removing the black frame.
A: The figures have been reworked.
- Secondary I also suggest to provide a figure with the chemical structure of the main identified compounds. This will help to guide the readers trough the manuscript.
A: The figure with the chemical structure of the main identified compounds has been added as Figure 3.
- As minor point, I would suggest to provide stars on the bar chart for significance.
A: The statistic in our document is based on Tukey's test and the result is various letters that indicate significance. We think that the use of asterisks to express significance is not appropriate in our case, given that e.g. in Figure 5 we have three different results simultaneously, indicated as lowercase, uppercase, and italics. Simultaneous use of asterisks and letters would be confusing for the reader. For this reason, we prefer to keep the expression of statistics as stated.
- It is unclear why and where MS was used in the manuscript. Was it used only for identification prior to NMR? This section is briefly mentioned in the material and methods I would kind ask to provide more details.
A: For better clarification of methodological steps and used methods, a flowchart was added as a figure 1. Because of higher sensitivity of MS detector, HPLC-MS was used for the determination of metabolites with detection limits under the limits of HPLC-DAD (Figure1). NMR identification was used just for structure confirmation of isolated 5-FQA.
- Additionally some parameters such as, spray voltage, mass isolation, calibrated lenses, declustering potential, are missing. I suggest to provide a table with this parameters in the supplementary material.
A: The required MS source data were supplemented as Table 2 in the section Materials and Methods.

Reviewer 2 Report
Dear authors - General comments,
- The manuscript entitled “Ethylene induction of non-enzymatic metabolic antioxidants in Matricaria chamomilla” is well organized and seems to be interesting. However, the topic of conclusion needs to be improved, because it was simply a repetition of the methodological steps and the results;
- I suggest that the authors make a schematic flowchart to facilitate the visualization of the methodological steps. Therefore, I request the construction of the flowchart to facilitate reading similar to the recently published article, see - Identification of Novel Chemical Entities for Adenosine Receptor Type 2A Using Molecular Modeling Approaches. Molecules 2020, 25(5), 1245;
- Figures need to improve resolution;
- Insert 2D structure of compounds showed in Figure 1. DAD-HPLC chromatogram of methanolic extract of Matricaria chamomilla leaves (320 nm); (1) skimmin, (2) daphnin, (3) chlorogenic acid, (4) Z-GMCA, (5) 5-FQA, (6) E-GMCA, (7) 1,5-DCQA, (8) herniarin.
Other comments
- The authors in the characterization of the compound do not show an important classical physical property to evaluate the purity of the compound (melting point);
- In the results and discussion there is no clear link between results and antioxidant capacity - It should be emphasized that the inhibition of enzymes or relate to "antioxidant activity". I request further clarification.
- I suggest that the authors include in silico predictions using the Swiss Target Prediction website (https://www.swisstargetprediction.ch) comparing these results with the molinspirtation server (https://www.molinspiration.com/) – I believe that the article would be more robust, since it is based on a very simple methodology to be carried out in a few minutes, and create an in topic Structure Activity Relationship;
- What was the criterion for the authors to perform only PAL activity and free radical assay (DPPH)? I suggest an adequate review of the literature for antioxidant activity and its application, add the cited reference in the manuscript - authors can consult the study to write similarly (An Antioxidant Potential, Quantum-Chemical and Molecular Docking Study of the Major Chemical Constituents Present in the Leaves of Curatella americana Linn (https://doi.org/10.3390/ph11030072). In addition, an additional study needs to be carried out (compounds) of chemical-quantum calculations similar to the published article by Costa, J.; Ramos, R.; Costa, K.; Brasil, D.; Silva, C.; Ferreira, E.; Borges, R.; Campos, J.; Macêdo, W.; Santos, C. An In Silico Study of the Antioxidant Ability for Two Caffeine Analogs Using Molecular Docking and Quantum Chemical Methods. Molecules 2018, 23, 2801. https://doi.org/10.3390/molecules23112801.
- The authors only claim that the molecule showed inhibition but did not elucidate the mechanism of action of the compound(s). An additional study of docking simulations or molecular dynamics are necessary, as mentioned in the articles previously;
- Only the molecular docking study informing distances and/or bond length is not enough to elucidate the mechanism of action. I suggest that additional molecular dynamics studies be conducted to investigate the molecular stability of the compounds always using a commercial compound (preferably the same as that used in biological assays in order to compare such in silico results with the experimental data). The authors' manuscript “worry me”, because during the experimental process there are several additional questions, such as the stability of the compound within the active site, as well as the pharmacokinetic and toxicological processes have not been carried out. Hence the importance of in silico predictions, as everything is mathematical models of predictions whether in vivo, in vitro or in silico.
- The experimental design is straightforward; the data are strong. However, such questions raised here need to be better clarified in order to be accepted. I firmly believe that the findings reported here will have a major impact in the scientific field, in my view is necessary to improve the manuscript and later I accept for publication.
Author Response
Reviewer 2
We thank Reviewer for incitement comments and comments. All suggested necessary changes were considered and are included in the manuscript.
- The manuscript entitled “Ethylene induction of non-enzymatic metabolic antioxidants in Matricaria chamomilla” is well organized and seems to be interesting. However, the topic of conclusion needs to be improved, because it was simply a repetition of the methodological steps and the results;
A: The conclusion has been rewritten.
- I suggest that the authors make a schematic flowchart to facilitate the visualization of the methodological steps. Therefore, I request the construction of the flowchart to facilitate reading similar to the recently published article, see - Identification of Novel Chemical Entities for Adenosine Receptor Type 2A Using Molecular Modeling Approaches. Molecules 2020, 25(5), 1245;
A: For better clarification of methodological steps and used methods, a flowchart was added as a Figure 1.
- Figures need to improve resolution;
A: The figures have been reworked.
- Insert 2D structure of compounds showed in Figure 1. DAD-HPLC chromatogram of methanolic extract of Matricaria chamomilla leaves (320 nm); (1) skimmin, (2) daphnin, (3) chlorogenic acid, (4) Z-GMCA, (5) 5-FQA, (6) E-GMCA, (7) 1,5-DCQA, (8) herniarin.
A: The figure with the chemical structure of the main identified compounds has been added as Figure 3.
- The authors in the characterization of the compound do not show an important classical physical property to evaluate the purity of the compound (melting point);
A: The required information (melting point 117-121 °C) of the compound has been added. The melting point for 5-FQA was determined from the same chromatographic fraction that was used for NMR structure identification and calibration on HPLC-DAD. The temperature range is given by the fact that it was not an absolutely pure fraction (negligible unidentifiable impurity).
- In the results and discussion there is no clear link between results and antioxidant capacity - It should be emphasized that the inhibition of enzymes or relate to "antioxidant activity". I request further clarification.
A: The Results and discussion section had been rewritten.
- I suggest that the authors include in silico predictions using the Swiss Target Prediction website (https://www.swisstargetprediction.ch) comparing these results with the molinspirtation server (https://www.molinspiration.com/) – I believe that the article would be more robust, since it is based on a very simple methodology to be carried out in a few minutes, and create an in topic Structure Activity Relationship;
- Data obtained from the recommended websites were incorporated into the manuscript – Table S1 in supplement material.
- What was the criterion for the authors to perform only PAL activity and free radical assay (DPPH)? I suggest an adequate review of the literature for antioxidant activity and its application, add the cited reference in the manuscript - authors can consult the study to write similarly (An Antioxidant Potential, Quantum-Chemical and Molecular Docking Study of the Major Chemical Constituents Present in the Leaves of Curatella americana Linn (https://doi.org/10.3390/ph11030072). In addition, an additional study needs to be carried out (compounds) of chemical-quantum calculations similar to the published article by Costa, J.; Ramos, R.; Costa, K.; Brasil, D.; Silva, C.; Ferreira, E.; Borges, R.; Campos, J.; Macêdo, W.; Santos, C. An In Silico Study of the Antioxidant Ability for Two Caffeine Analogs Using Molecular Docking and Quantum Chemical Methods. Molecules 2018, 23, 2801. https://doi.org/10.3390/molecules23112801.
- The authors only claim that the molecule showed inhibition but did not elucidate the mechanism of action of the compound(s). An additional study of docking simulations or molecular dynamics are necessary, as mentioned in the articles previously;
- Only the molecular docking study informing distances and/or bond length is not enough to elucidate the mechanism of action. I suggest that additional molecular dynamics studies be conducted to investigate the molecular stability of the compounds always using a commercial compound (preferably the same as that used in biological assays in order to compare such in silico results with the experimental data). The authors' manuscript “worry me”, because during the experimental process there are several additional questions, such as the stability of the compound within the active site, as well as the pharmacokinetic and toxicological processes have not been carried out. Hence the importance of in silico predictions, as everything is mathematical models of predictions whether in vivo, in vitro or in silico.
A: We thank the opponent for inspiring comments and pointing out the possibilities of theoretical prediction of the antioxidant effect based on the quantum calculations. Based on the recommended publications, we partially processed molecular modelling for the three studied quinic derivatives. We incorporated into the manuscript data obtained from mapping of molecular electrostatic potential (MEP) and the frontier orbitals (HOMO and LUMO). Molecular docking is a good tool for the theoretical evaluation of the possible mechanism of action of potential antioxidants in the prevention of ROS formation. However, this method is prolonged and requires more detailed analysis in the selection of target substances (enzymes, proteins, etc.). Our main goal was to monitor the changes in the content of the substances due to exogenously applied ethephon. Docking will be the subject of a detailed study, which we will extend to the potential effects of other substances with increased accumulation after the application of exogenous stimuli.
- The experimental design is straightforward; the data are strong. However, such questions raised here need to be better clarified in order to be accepted. I firmly believe that the findings reported here will have a major impact in the scientific field, in my view is necessary to improve the manuscript and later I accept for publication.

Reviewer 3 Report
The manuscript in reference describes the effect of the plant growth regulator ethephon (2-chloroethyl)phosphonic acid) on the content of some phenolic compounds in young tetraploid chamomile plants. The manuscript is interesting and contains important information and results, even though some sections are very confusing. Therefore, some points need to be addressed before further consideration.
- A detailed scrutiny should be carried out throughout the manuscript to review and correct some stylistic and grammatical inaccuracies, such as lines 20,37,49,55,65,69,87,97,124,171, among several others.
- The results and discussion section seems to be cluttered. It is very difficult to follow the direction of ideas. I recommend subdividing this section into subheadings including the different types of results involved, to focus readers on the goal of each part of this section. A scheme explaining the different parts is also recommended.
- Unify the expression "non-enzymatic" to divide it with hyphens or not.
- Line 79: check MS/DAD-HPLC, as it should be HPLC-DAD-MS.
- Regarding the HPLC-DAD-MS analyses, no information on the MS characteristics of the compounds is discussed in the manuscript, and is only mentioned in Materials and Methods.
- Clarify in the manuscript which sample is represented by the chromatogram in Figure 1. I mean, is it of control, ET-treated at 24 h, etc?.
- Clarify, justify and explain in the manuscript why a 0.01% ET solution was used and other ET concentrations were not tested.
- Specify the origin, brand (if applicable), and quality of the standards used for quantitative analysis in Table 1.
- Figures 1, 2, 3, 5, 6, A1, A2 do not have captions on the X and/or Y axes.
- Table 1: Clarify in the manuscript if the control is related to time = 0 h or is related to plants without ET treatment. Unify this with other figures (for instance, Figure 2). In this sense, if the control is related to time = 0 h, why weren't untreated plants included in the experiment to evaluate the accumulation over time of some phenolics as a control? This musst be clarified and explaind in the manuscript.
Author Response
Reviewer 3
We thank Reviewer for incitement comments and comments. All suggested necessary changes were considered and are included in the manuscript.
- A detailed scrutiny should be carried out throughout the manuscript to review and correct some stylistic and grammatical inaccuracies, such as lines 20,37,49,55,65,69,87,97,124,171, among several others.
A: The manuscript was checked for grammatical and stylistic errors.
- The results and discussion section seems to be cluttered. It is very difficult to follow the direction of ideas. I recommend subdividing this section into subheadings including the different types of results involved, to focus readers on the goal of each part of this section. A scheme explaining the different parts is also recommended.
A: The section Results and discussion has been reworked and devided into subheadings. For better clarification of methodological steps and used methods, a flowchart was added as a Figure 1.
- Unify the expression "non-enzymatic" to divide it with hyphens or not.
A: The expression was unified as non-enzymatic.
- Line 79: check MS/DAD-HPLC, as it should be HPLC-DAD-MS.
A: It has been rewritten.
- Regarding the HPLC-DAD-MS analyses, no information on the MS characteristics of the compounds is discussed in the manuscript, and is only mentioned in Materials and Methods.
A: The MS method was used for determination of phenolic acids and flavonoids. These data are easily accessible so we did not list them in more detail.
- Clarify in the manuscript which sample is represented by the chromatogram in Figure 1. I mean, is it of control, ET-treated at 24 h, etc?.
A: The legend for Figure 2 has been modified to clearly explain the origin of the chromatogram.
- Clarify, justify and explain in the manuscript why a 0.01% ET solution was used and other ET concentrations were not tested.
A: The results of the work of Sajko et al. 2018 (DOI 10.1007/s00344-017-9735-1) devoted to the effect of 0.01% ethephon on chamomile plants point to significant changes in the accumulation of some coumarins, total soluble phenols and flavonoids. However, the work does not specifically address phenolic acids or flavonoids, which are more effective antioxidants. For this reason, we chose the given concentration and focused in detail on the mentioned metabolites.
- Specify the origin, brand (if applicable), and quality of the standards used for quantitative analysis in Table 1.
A: The required information of the compound has been added in section Materials and Methods.
- Figures 1, 2, 3, 5, 6, A1, A2 do not have captions on the X and/or Y axes.
A: The figures have been reworked and all necessary data for the x and y axes have been added.
- Table 1: Clarify in the manuscript if the control is related to time = 0 h or is related to plants without ET treatment. Unify this with other figures (for instance, Figure 2). In this sense, if the control is related to time = 0 h, why weren't untreated plants included in the experiment to evaluate the accumulation over time of some phenolics as a control? This musst be clarified and explaind in the manuscript.
A: In all used figures, tables, or in the text, the writing of the control group was unified, as the time 0 h.

Round 2
Reviewer 2 Report
Dear authors, The manuscript has improved significantly, but I request minor comments before the publication. 1. In line 383 5-FQA (-5,5648 eV) - replace comma for point; 2. In Figure 1S and I suggest adding the unit (a.u and eV); 3. In item 3.6. Molecular modelling - The authors do not inform all requested in silicos procedures (only quantum chemical calculations); 4. In lines 392-403 - I didn't see any discussion of the results just a description without any comparison with the available experimental data from the literature, for example, with the data obtained in this study, 5. 1,5-DCQA has molecular mass (516.46), violating a rule for oral bioavailability. The authors do not discuss the importance of rule violation and the influence of the logP property in the study, for example. 6. Images (2-8) need to be improved.Author Response
We would like to thank the Reviewer for valuable comments that helped us to improve the manuscript. In following we list the changes made in the paper according to recommended revisions.
